# SimpleTIR: End-to-End Reinforcement Learning for Multi-Turn Tool-Integrated Reasoning

**Zhenghai Xue**[1*]    **Longtao Zheng**[1*]    **Qian Liu**[2†]    **Yingru Li**[2]
**Xiaosen Zheng**[2]    **Zejun Ma**[2]    **Bo An**[1]
[1]Nanyang Technological University, Singapore    [2]TikTok, Singapore

## Abstract

Large Language Models (LLMs) can significantly improve their reasoning capabilities by interacting with external tools, a paradigm known as Tool-Integrated Reasoning (TIR). However, extending TIR to multi-turn scenarios using Reinforcement Learning (RL) is often hindered by training instability and performance collapse. We identify that such instability is primarily caused by a distributional drift from external tool feedback, leading to the generation of extremely low-probability tokens. This issue compounds over successive turns, causing erroneous responses and catastrophic gradient norm explosions. Through careful inspections, we find a phenomenon named "void turns" as a preferable sign of pathological generations, where neither a code block nor a final answer is yielded. We introduce SimpleTIR, a plug-and-play algorithm that identifies and filters out trajectories containing void turns when performing the policy update. SimpleTIR effectively blocks the harmful, high-magnitude gradients, thus stabilizing the learning dynamics. Extensive experiments show that SimpleTIR achieves state-of-the-art performance on challenging math reasoning benchmarks, notably elevating the AIME24 score from a text-only baseline of 22.1 to 50.5 when starting from the Qwen2.5-7B base model. Furthermore, by avoiding the constraints of supervised fine-tuning, SimpleTIR encourages the model to discover diverse and sophisticated reasoning patterns, such as self-correction and cross-validation.

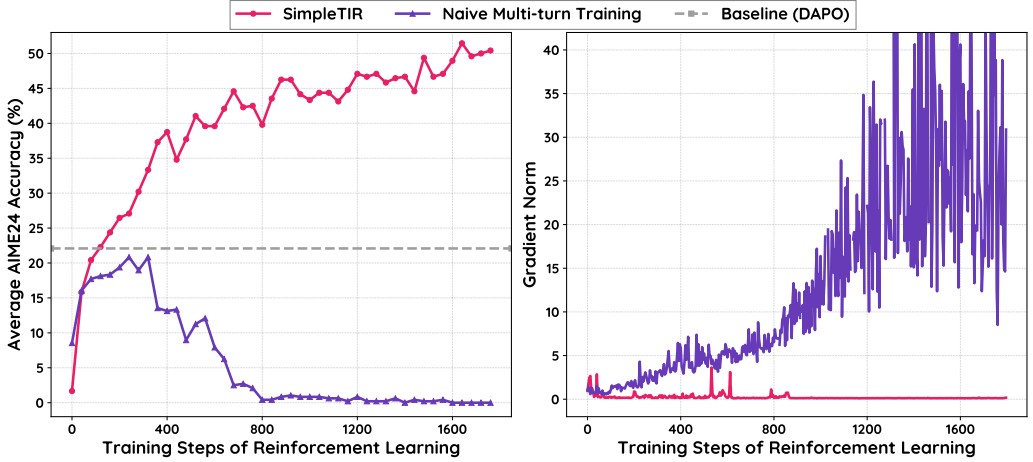

**Figure 1:** Starting from Qwen2.5-7B base model, The training dynamics of SimpleTIR are highly stable, and it clearly outperforms the baseline method without TIR (DAPO). The gradient norm remains well-behaved with almost no spikes. In contrast, Naive Multi-turn Training not only suffers from unstable dynamics and catastrophic gradient norm explosions, but also fails to match the performance of the baseline without TIR.

---

*The first two authors contribute equally.
†Correspondence: Qian Liu (liuqian.sea@gmail.com)

# 1 INTRODUCTION

Training Large Language Models (LLMs) from pretrained models using only outcome-based rewards, a paradigm known as Zero Reinforcement Learning (RL), represents a fundamental step towards autonomous agents and has garnered significant attention since the success of DeepSeek-R1 (DeepSeek-AI Team, 2025). By freeing models from carefully selected Supervised Fine-tuning (SFT) data, Zero RL holds the potential to unlock more general problem-solving capabilities and foster the emergence of diverse, novel reasoning patterns. A critical frontier for realizing this paradigm's potential is enabling LLMs to perform multi-turn Tool-Integrated Reasoning (TIR), where models iteratively reason, generate code, execute it, and use the output for informed reasoning in subsequent turns. TIR directly addresses inherent LLM limitations such as poor computational accuracy and knowledge cutoffs by allowing them to leverage external tools like a Python interpreter or a search engine. Despite its clear potential, training LLMs for multi-turn TIR with Zero RL remains highly challenging due to frequent instability and gradient explosion issues (Liu et al., 2025a; Mai et al., 2025; Baronio et al., 2025; Moonshot AI, 2025; Li et al., 2025c). One common solution is to "cold start" the model with distilled TIR trajectories to enhance stability (Feng et al., 2025). However, this SFT-based approach fundamentally violates the core philosophy of Zero RL, constraining the model to human-annotated patterns and thus preventing the discovery of novel reasoning strategies.

In this paper, our key insight lies in identifying the root cause of this instability: the emergence and accumulation of extremely *low-probability tokens* stemming from a distributional drift. When external tool feedback is used as model input in multi-turn TIR, such input may deviate from the model's pretrained data distribution. Although the tool feedback itself is masked when computing the policy loss (Jin et al., 2025; Mai et al., 2025), the model's subsequent generations inherit this distributional shift, leading to increased stochasticity. Consequently, the model is more likely to sample low-probability tokens. This issue is compounded in the multi-turn loop, as these low-probability tokens are fed back as input, exacerbating the distributional shift in subsequent turns. We make theoretical analysis of the gradient norm on softmax logits and find two dominating terms with negative correlations to token probabilities. This explains the gradient explosion issues observed in prior work.

Building on this analysis, we propose SimpleTIR, an effective trajectory filtering algorithm that stabilizes multi-turn TIR training. We observe that the accumulation of low-probability tokens and high generation stochasticity frequently results in what we define as a *void turn*: an LLM response that contains neither a complete code block nor a final answer. It serves as a powerful heuristic indicator for identifying problematic trajectories. The core strategy of SimpleTIR is to filter out trajectories containing void turns. By excluding these trajectories from the policy loss computation, SimpleTIR blocks the harmful, high-magnitude gradients associated with the problematic low-probability sequences, directly addressing the gradient explosion issue. This filtering approach is general and plug-and-play, making it a practical solution for stabilizing Zero RL training in different scenarios without significant engineering overhead.

To demonstrate the effectiveness of SimpleTIR, we conduct comprehensive experiments on challenging mathematical reasoning tasks. SimpleTIR establishes a new state-of-the-art on a suite of challenging mathematical reasoning benchmarks, improving the AIME24 score from 22.1 to 50.5 on Qwen2.5-7B and outperforming prior methods trained on specialized data. Our ablation studies confirm that filtering trajectories with void turns is the crucial component for stabilizing training, overcoming the instability that plagues naive multi-turn approaches and enabling significant performance gains. We also highlight a key advantage of the Zero RL approach. In contrast to methods that rely on a "cold-start" SFT phase, SimpleTIR encourages the model to discover novel and diverse reasoning patterns such as cross-validation, progressive reasoning, and self-correction, facilitating future data-based researches.

# 2 PRELIMINARIES

End-to-end training of multi-turn TIR agents with RL is challenging due to its compositional structure. We therefore model the process as a Hierarchical Markov Decision Process (Hauskrecht et al., 1998), which separates decision-making into two levels: a high-level policy governing the sequence of conversational turns and a low-level policy for generating tokens within each turn.

## 2.1 Hierarchical MDP Formulation for Multi-Turn TIR

A full interaction trajectory is a sequence $o = (q, l_0, f_0, \ldots, l_{K-1}, f_{K-1})$, where $l_k$ is the model's generated response at turn $k$ and $f_k$ is the subsequent tool feedback. The high-level MDP, $\mathcal{M}_H = \langle \mathcal{S}_H, \mathcal{A}_H, T_H, R_H, \gamma_H \rangle$, operates at the turn level to govern the overall strategy. It includes the state $S_k = (q, l_0, f_0, \ldots, l_{k-1}, f_{k-1})$ representing the complete conversation history before the current turn, the action $A_k$ indicating an option, or high-level sub-policy, the transition $T_H: S_{k+1} = S_k \circ (l_k, f_k)$, and the terminal reward assigned to the final trajectory based on its overall success. The low-level MDP, $\mathcal{M}_L = \langle \mathcal{S}_L, \mathcal{A}_L, T_L, R_L, \gamma_L \rangle$, operates at the token level to execute the chosen high-level action (Li et al., 2024), including the state $s_t = S_k \circ (a_1, \ldots, a_{t-1})$ indicating the sequence of tokens generated so far within the current turn $k$, the action $a_t \in \mathcal{A}_L$ which is a single token selected from the model's vocabulary, the transition $T_L: s_{t+1} = s_t \circ a_t$, and the reward $R_L = 0$. The low-level policy receives no intrinsic reward, as its only goal is to complete the high-level action.

In this framework, $\circ$ denotes concatenation, and we set the discount factor $\gamma = \gamma_H = \gamma_L = 1$. We train a single, unified policy $\pi_\theta(a_t | s_t)$ to implicitly solve this two-level problem.

## 2.2 Joint Policy Optimization and Feedback Masking

The unified policy $\pi_\theta$ is trained using Group Relative Policy Optimization (GRPO) (DeepSeek-AI Team, 2025). GRPO circumvents the need for a learned value function (Li et al., 2024) by calculating the advantage based on the relative performance within a group of $G$ trajectories sampled from the same prompt. The advantage for trajectory $o_i$ is $\hat{A}_i = r_i - \text{mean}\left(\{r_j\}_{j=1}^G\right)$, where $r_i$ is the terminal reward from the high-level MDP.

A critical adaptation is required for the TIR setting. The policy is only responsible for generating response tokens ($l_k$), not the environment-provided feedback tokens ($f_k$). To ensure correct credit assignment, we employ feedback token masking (Jin et al., 2025; Mai et al., 2025). The loss is accumulated only over the timesteps corresponding to the agent's actions, effectively excluding feedback tokens from the gradient computation.

This leads to our final training objective, $\mathcal{J}_{\text{TIR}}(\theta)$:

$$\mathcal{J}_{\text{TIR}}(\theta) = \mathbb{E}_{\substack{q \sim q_0, \\ \{o_i\}_{i=1}^G \sim \pi_{\theta_{\text{old}}}(\cdot | q)}} \left[ \frac{1}{G} \sum_{i=1}^G \frac{1}{\sum_t m_{i,t}} \sum_{t=1}^{|o_i|} m_{i,t} \cdot L_{\text{CLIP}}(\theta, i, t) \right], \quad (1)$$

where $m_{i,t}$ is a binary mask that is 1 if the token at step $t$ belongs to any response $l_k$ and 0 otherwise. The term $L_{\text{CLIP}}$ is the standard clipped surrogate objective from PPO (Schulman et al., 2017; Wang et al., 2019): $L_{\text{CLIP}}(\theta, i, t) = \min\left(\rho_{i,t}(\theta)\hat{A}_i, \text{clip}(\rho_{i,t}(\theta), 1 - \varepsilon, 1 + \varepsilon)\hat{A}_i\right)$, with the importance sampling ratio $\rho_{i,t}(\theta) = \frac{\pi_\theta(o_{i,t} | o_{i,<t})}{\pi_{\theta_{\text{old}}}(o_{i,t} | o_{i,<t})}$.

## 3 Methodology

In this section, we first identify a certain abnormal model behavior specific to multi-turn TIR: the emergence of low-probability tokens. We further interconnect such behavior with the phenomenon of unstable multi-turn TIR training. It is demonstrated how these tokens drive gradient explosions and create a credit assignment dilemma during Zero-RL training. To alleviate the influence of low-probability tokens, we propose an effective trajectory filtering approach, leading to the SimpleTIR algorithm that stabilizes multi-turn training and ensures that successful early turns are not penalized for a later collapse.

### 3.1 The Emergence of Low-Probability Tokens in Multi-Turn TIR

The instability of multi-turn TIR training is a known challenge (Wang et al., 2025; Mai et al., 2025), yet its root cause remains largely underexplored. Instability can plausibly arise from two stages: the choice of using tools (before tool calling) and the follow-up analysis of tool feedback (after tool calling). We contrast multi-turn TIR with a minimal, single-turn TIR setting where the model produces exactly one response containing reasoning and an optional code block. This single-turn

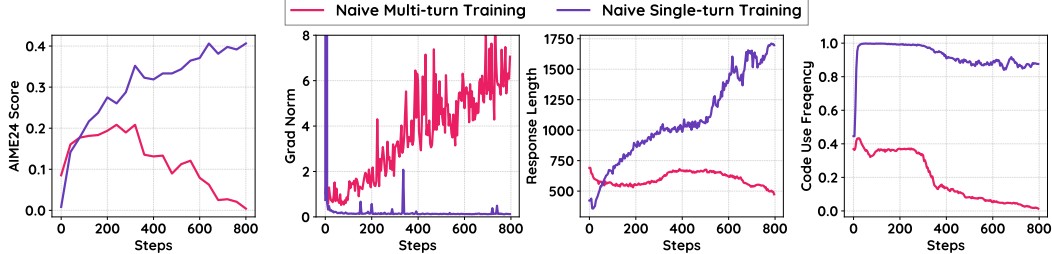

**Figure 2:** Training statistics comparing naive single-turn and multi-turn TIR. Single-turn training proceeds smoothly and achieves higher performance, while multi-turn training is unstable.

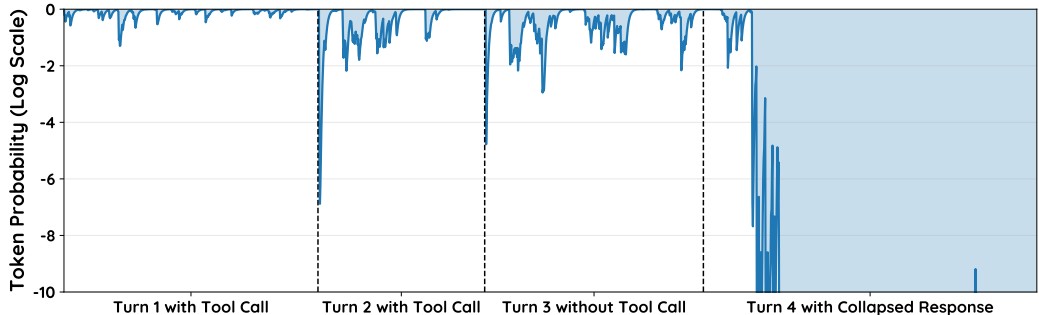

**Figure 3:** Visualization of token probabilities in a multi-turn TIR trajectory. The y-axis is log-scaled. Distributional drift from tool feedback in early turns leads to a collapse in token probabilities in later turns. The detailed responses are in Tab. 4 of Appendix A.1.

setup involves no analysis of tool feedback, thereby isolating the pre-tool-call reasoning phase. As shown in Fig. 2, the single-turn baseline trains smoothly and achieves reasonable performance, whereas the multi-turn equivalent suffers from performance collapse and recurring gradient spikes.

This comparative analysis indicates that the key factor contributing to instability lies in the tool feedback loop. In multi-turn TIR, the tool feedback $f_k$ from turn $k$ is concatenated into the prompt for turn $k + 1$. Because this feedback originates from an external interpreter, it can deviate significantly from the LLM's pretrained data distribution in Zero RL. Conditioned on such out-of-distribution (OOD) input, the model's subsequent generations can drift away from pretrained patterns, becoming highly stochastic and assigning anomalously low probabilities to selected tokens.

We verify this claim with a case study in Fig. 3. We train the multi-turn TIR with the original GRPO algorithm for 20 steps and select a trajectory with erroneous responses[1]. According to the figure, the tool feedback after Turn 1 and Turn 2 contains tokens with extremely low probabilities, confirming their OOD nature. While masking the loss on these feedback tokens is a known practice (Jin et al., 2025; Mai et al., 2025), it is insufficient. The distributional drift induced by these tokens contaminates subsequent model generations. As seen in Fig. 3, while response token probabilities in Turn 1 are relatively high, low-probability segments emerge in the model's own text in Turns 2 and 3. This compounding drift culminates in a collapsed, nonsensical response with extremely low token probabilities in Turn 4.

## 3.2 HOW LOW-PROBABILITY TOKENS COMPROMISE ZERO-RL TRAINING

After identifying the emergence of low-probability tokens, we analyze their two primary detrimental effects on Zero-RL training: gradient explosion and misaligned credit assignment.

**Gradient Explosion.** A primary failure mode in multi-turn TIR is the explosion of gradient norms (Fig. 2). Intuitively, as the token probability $\pi_{\text{old}}$ appears in the denominator of the PPO ratio term,

---

[1]Detailed responses in this turn can be found in Tab. 4 in Appendix A.1

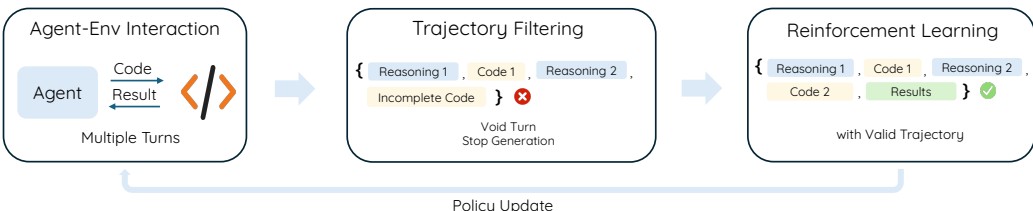

**Figure 4:** An overview of SIMPLETIR. During the policy update, SIMPLETIR identifies and filters out entire trajectories that contain a *void turn*—an LLM response that fails to produce either a complete code block or a final answer.

low-probability tokens can directly lead to the high policy gradients. To formalize this, we analyze the policy gradient with respect to the pre-softmax logits **z** following Li (2025). This gradient is component of the total gradient with respect to model parameters.

**Proposition 1.** *Consider a token $c$ at timestep $t$ of a trajectory $o_i$. The L2 norm of the policy gradient with respect to the logits $\mathbf{z}_t$ is:*

$$\|\nabla_{\mathbf{z}_t}\mathcal{J}_{TIR}\|_2 = \frac{m_{i,t}}{\sum_j m_{i,j}} \cdot \rho_{i,t}(\theta) \cdot g_{i,t} \cdot |\hat{A}_i| \cdot \sqrt{1 - 2P(c) + \sum_{j\in\mathcal{A}} P(j)^2}, \qquad (2)$$

*where $m_{i,t}$ is the feedback mask, $\rho_{i,t}(\theta)$ is the importance ratio, $|\hat{A}_i|$ is the absolute advantage, $P$ is the policy's probability distribution $\pi_\theta(\cdot|o_{i,<t})$, and $g_{i,t}$ is a gating function active when the PPO update is not clipped.*

Proposition 1 reveals that the gradient norm is highly sensitive to two factors that are exacerbated by low-probability tokens:

- **Unclipped Importance Ratio:** The ratio $\rho_{i,t}(\theta) = \frac{\pi_\theta(o_{i,t}|o_{i,<t})}{\pi_{\theta_{old}}(o_{i,t}|o_{i,<t})}$ is a primary source of gradient spikes. For a negatively-rewarded trajectory ($\hat{A}_i < 0$), this ratio is unbounded from above. If token $c$ was generated with a very low probability by the old policy, $\pi_{\theta_{old}}(c|\cdot)$ is minute. Even a small update to $\pi_\theta(c|\cdot)$ can cause $\rho_{i,t}(\theta)$ to explode, leading to the gradient spikes observed in training.

- **Sustained High Gradient Norm:** The probability-dependent term, $\sqrt{1 - 2P(c) + \sum_j P(j)^2}$, can sustain large gradients. When the policy assigns a low probability to the sampled token $c$, $1 - 2P(c)$ nears its maximum of 1. If the policy is otherwise confident (i.e., the distribution is sharp), the collision probability $\sum_j P(j)^2$ remains large, preventing the gradient norm from diminishing and thereby contributing to unstable training (Li, 2025).

**Misaligned Credit Assignment.** Beyond gradient instability, low-probability tokens introduce a severe credit assignment problem. As seen in Fig. 3, these tokens are more prevalent in later turns. With a sparse, terminal reward, a trajectory that fails in its final turns receives a single negative reward for the entire sequence. This signal does not distinguish between correct, high-probability reasoning in early turns and the faulty, low-probability tokens that caused the eventual failure. This dynamic unfairly penalizes valid multi-turn behavior, causing the policy to collapse toward safer, single-turn generations.

### 3.3 SimpleTIR: Stabilizing Training by Filtering Void Turns

Given that low-probability tokens are the root cause, simple heuristics like masking high-perplexity trajectories or clipping the importance ratio may seem appealing. While effective in some contexts (Zheng et al., 2025a; Zhang et al., 2025), we show in Fig. 5 (bottom) that these methods fail to resolve instability in multi-turn TIR, as their thresholds are difficult to tune and they do not adequately address the credit assignment problem.

A more robust filtering criterion is needed. We observe that the collapsed Turn 4 in Fig. 3 follows a turn that produced *neither a tool call nor a final answer*. Intuitively, such a turn makes no progress in

the reasoning process and should not appear in an effective TIR trajectory. We define these turns as **void turns**, with incomplete code blocks being common causes. Concrete examples of trajectories containing void turns can be found in Appendix A.1. Void turns are often symptoms of distributional drift, where high generation stochasticity leads to a premature end-of-sequence token. Because void turns are rare in successful trajectories but indicative of pathological ones, they serve as a powerful heuristic for identifying problematic trajectories.

This insight leads to the SimpleTIR algorithm, illustrated in Fig. 4. The procedure is straightforward: for each sampled trajectory, we inspect its turns. If any turn contains neither a complete code block nor a final answer, it is labeled a void turn. We then mask the policy loss for the entire trajectory, removing it from the batch before the GRPO update. This single step simultaneously prevents the large gradients from low-probability tokens from backpropagating and corrects misaligned credit assignment by ensuring successful early turns are not penalized for a later collapse. SIMPLETIR's filtering approach is agnostic to the specific RL algorithm used and is orthogonal to other recent improvements in RL for LLM reasoning (Liu et al., 2025a; Zheng et al., 2025a; Chen et al., 2025; Li et al., 2025c; 2026b;a).

### 3.4 IMPLEMENTATION DETAILS

To further enhance training stability and efficiency, we adopt several key practices. First, to avoid out-of-distribution special tokens when using base models, we do not use chat templates. Instead, we prepend tool outputs with a simple prefix, "Code Execution Result:". Second, to provide a shortcut for simple tasks and improve sample efficiency, we prepend every LLM-generated code block with a 'final_answer' function, allowing the model to terminate and answer within a single turn if possible. Finally, to prevent the model from hallucinating tool outputs, we strictly halt LLM generation after a complete code block is produced and always append the true, external tool feedback before the next turn begins.

## 4 EXPERIMENTS

### 4.1 SETUP

**Training**   We prepare our training code with the VeRL (Sheng et al., 2024) and Search-R1 (Jin et al., 2025) framework. We use Sandbox Fusion as an asynchronous code interpreter. The training datasets are Math3-5 from SimpleRL (Zeng et al., 2025) and Deepscaler (Luo et al., 2025). SimpleTIR follows the Zero RL setting and uses the unaligned Qwen series as the base models, including Qwen2.5-7B, Qwen2.5-32B, and Qwen3-4B-Base. During training, the rollout batch size is set to 512, and the mini update size is set to 128. The maximum response length is initially set to 16K, with a maximum of five turns of code execution. When the average response length plateaus, we increase the maximum response length to 24K and the largest number of turns to 10. Other training hyperparameters are in Appendix B.2.

**Evaluation**   Our evaluation is conducted on Math500 (Hendrycks et al., 2021), AIME24, AIME25, AMC23, and Hmmt Feb 25, using a temperature of 1 and reporting average@32 scores to reduce variance, following the standard practice in evaluating LLMs' reasoning abilities (Yu et al., 2025). For comparison, we consider three categories of baselines. The first is non-TIR Zero RL, where we use SimpleRL-Zoo (Zeng et al., 2025) and DAPO (Yu et al., 2025) as representative baselines. The score of 22.1 in Fig. 1 is exactly obtained with DAPO on the Qwen2.5-7B model. The performance gap between these methods and SimpleTIR highlights the advantage of incorporating TIR in mathematical reasoning. The second category is TIR RL from cold-start or specialized models. ARPO (Dong et al., 2025) finetunes Qwen2.5-7B-Instruct. ToRL (Li et al., 2025b), VerlTool (Jiang et al., 2025), and Effective CIR (Bai et al., 2025) apply RL to the Qwen2.5-Math series. The final category is Zero RL with TIR, where, to the best of our knowledge, Zero-TIR (Mai et al., 2025) is the only method that strictly follows the Zero RL paradigm by training TIR models directly from base models. To the best of our knowledge, there is no published work that finetunes the Qwen3-4B-Base model with RL. So for SimpleTIR-4B we only prepare it with the base model.

**Table 1:** Performance comparison on various math benchmarks. Check and cross marks in the "TIR" column refers to whether the method involves TIR during training and evaluation. The "From" column indicates the type of base models we train the model from. We fill the scores with - if they are not provided in respective reports.

| Model | TIR | From | AIME24 | AIME25 | MATH500 | Olympiad | AMC23 | Hmmt25 |
|-------|-----|------|--------|--------|---------|----------|-------|--------|
| *Models based on Qwen2.5-7B* | | | | | | | | |
| Qwen2.5-7B | ✗ | Base | 3.2 | 1.1 | 51.9 | 15.4 | 21.7 | 0.0 |
| Qwen2.5-7B-TIR | ✓ | Base | 1.7 | 0.6 | 18.0 | 6.2 | 10.8 | 1.9 |
| SimpleRL-Zoo-7B | ✗ | Base | 15.6 | - | 78.2 | 40.4 | 62.5 | - |
| ToRL-7B | ✓ | Math-Inst | 40.2 | 27.9 | 82.2 | 49.9 | 75.0 | - |
| Effective TIR-7B | ✓ | Math | 42.3 | 29.2 | 86.4 | - | 74.2 | - |
| VT-7B | ✓ | Math | 36.7 | - | 82.8 | 51.6 | 75.0 | - |
| ARPO-7B | ✓ | Inst | 30.0 | 30.0 | 78.8 | - | - | - |
| ZeroTIR-7B | ✓ | Base | 39.6 | 25.0 | 80.2 | - | - | 22.5 |
| SimpleTIR-7B | ✓ | Base | **50.5** | **30.9** | **88.4** | **54.8** | **79.1** | **29.7** |
| *Models based on Qwen2.5-32B* | | | | | | | | |
| Qwen2.5-32B | ✗ | Base | 4.2 | 1.6 | 43.1 | 17.8 | 28.0 | 0.2 |
| Qwen2.5-32B-TIR | ✓ | Base | 7.1 | 5.0 | 37.0 | 16.9 | 20.0 | 5.2 |
| DAPO | ✗ | Base | 50.0 | - | - | - | - | - |
| ZeroTIR-32B | ✓ | Base | 48 | 27 | 87.8 | - | - | 20.0 |
| SimpleTIR-32B | ✓ | Base | **59.9** | **49.2** | **92.9** | **63.7** | **91.6** | **34.6** |
| *Models based on Qwen3-4B* | | | | | | | | |
| Qwen3-4B-Base | ✗ | Base | 4.9 | 2.9 | 46.6 | 18.1 | 21.7 | 0.5 |
| Qwen3-4B-TIR | ✓ | Base | 4.9 | 3.1 | 24.9 | 12.7 | 15.9 | 2.9 |
| SimpleTIR-4B | ✓ | Base | **48.1** | **40.2** | **90.0** | **57.4** | **83.1** | **28.2** |

## 4.2 TRAINING RESULTS

As shown in Tab. 1, SimpleTIR achieves significant performance improvements over the base models and outperforms all Zero RL baselines, both with and without TIR. Notably, SimpleTIR also surpasses baselines that start from specialized, math-instructed models, such as ToRL and Effective TIR. This result highlights the power and potential of the Zero RL paradigm: by ensuring training stability and avoiding the premature penalization of complex multi-turn behaviors, models can develop sophisticated reasoning abilities directly from outcome-based rewards, without relying on human-annotated SFT data. Furthermore, the consistent performance gains observed across various base models, including Qwen2.5-7B, Qwen2.5-32B, and Qwen3-4B-Base, confirm the general applicability and scalability of the SimpleTIR framework.

## 4.3 TRAINING CURVES, METRICS, AND ABLATION STUDIES

We show the training curve of SimpleTIR from Qwen2.5-7B with 1, 5, and 10 turns of generation in Fig. 5 (Top). In all these settings, SimpleTIR exhibits constant and smooth increases of the average response length and performance scores. The average number of turns first arises quickly then remains constant for multi-turn SimpleTIR. We also observe that the response length and the Math500 score scales with more turns, while the AIME24 score does not benefit clearly. This indicates that different tasks require distinct reasoning patterns. Some may be solvable with few steps of reasoning, but others will take a number of external feedback before reaching the correct answer.

We also conduct ablation studies to demonstrate the effectiveness of trajectory filtering in SimpleTIR. We first investigate two alternative filtering criteria: high importance ratio and low token probabilities, as specified in the first paragraph of Sec. 3.3. As shown in Fig. 5 (Bottom), these two filtering approach cannot resolve the issue of gradient explosion, exhibiting unstable curves of training scores. SimpleTIR features a more stable curve of gradient norm, thanks to the mild token probability distributions. This demonstrates the effectiveness of void turn filtering in stabilizing multi-turn TIR training. We then consider an ablation method where LLM generation is terminated on void turns

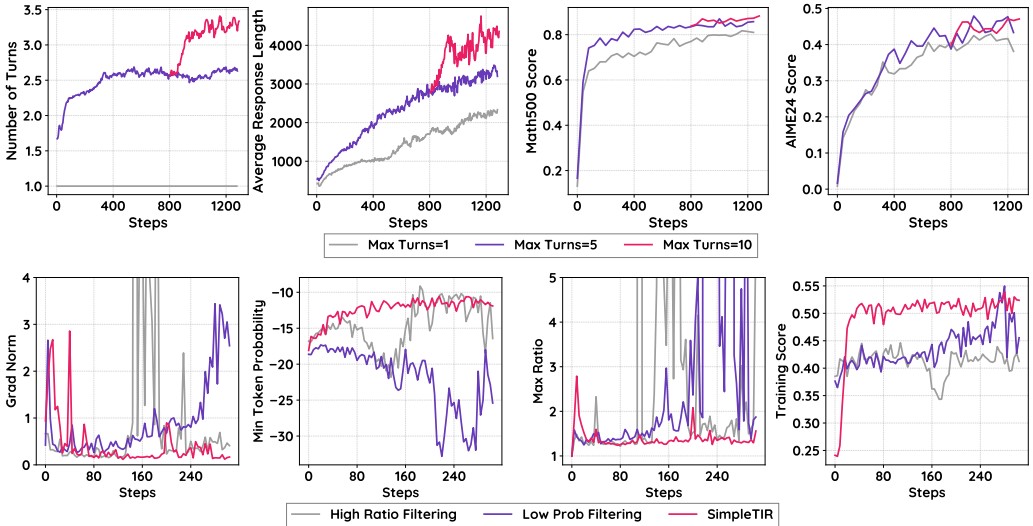

**Figure 5: Top**: Training curves for SimpleTIR with different maximum number of turns. SimpleTIR with maximum 10 turns is resumed at 200 steps from SimpleTIR with maximum 5 turns. SimpleTIR clearly benefits from scaling interaction turns from 1 to 5. **Bottom**: The training curves for ablation studies in the first 320 steps. Trajectory filtering with high importance ratios or low probability tokens cannot resolve the challenge of training instability, while SimpleTIR suffers less from low probability tokens and gradient explosion.

**Table 2:** Results of ablation studies. Considering the unstable training of ablated methods, we report the highest scores within 1000 gradient steps. "Naive Multi-Turn" directly applies RLVR in multi-turn TIR. "Low Prob" and "High Ratio" filtering refers to masking the policy loss on tokens with lowest probabilities or highest importance ratio.

|          | SimpleTIR-7B | Naive Multi-Turn | Low Prob Filtering | High Ratio Filtering | Stop Generation w/o Filtering |
|----------|--------------|------------------|--------------------|----------------------|-------------------------------|
| AIME24   | **50.5**     | 20.8             | 23.3               | 26.3                 | 26.1                          |
| Math500  | **88.4**     | 73.1             | 72.8               | 75.0                 | 77.3                          |

but resulting trajectories are not filtered when computing policy loss. According to the validation results in Tab. 2, this method is also inferior to SimpleTIR. This can be attributed to misaligned credit assignment since trajectories containing void turns can hardly obtain positive outcome. SimpleTIR handles such issue by masking the loss of whole responses containing void turns.

## 4.4 EMERGENCE OF DIVERSE REASONING BEHAVIORS

Thanks to the framework of Zero RL training, SimpleTIR automatically reinforces useful reasoning patterns obtained in the pretraining phase, rather than sticking to predefined patterns in the SFT dataset. In Appendix A.2, we show SimpleTIR responses with diverse multi-turn reasoning behaviors. They are mostly combinations of the three main reasoning patterns illustrated in Figure 6, namely Cross Validation, Progressive Reasoning, and Error Correction.

We also use Claude-3.7-Sonnet to identify and count the frequency of reasoning patterns in responses generated by ReTool and SimpleTIR-32B. The responses are filtered so that they all lead to the correct final answer. Both models demonstrate a strong tendency to conduct multiple rounds of cross verification. Meanwhile, SimpleTIR-32B exhibits more instances of progressive reasoning and error correction. This illustrates the advantage of Zero RL, preserving more diversity in reasoning patterns.

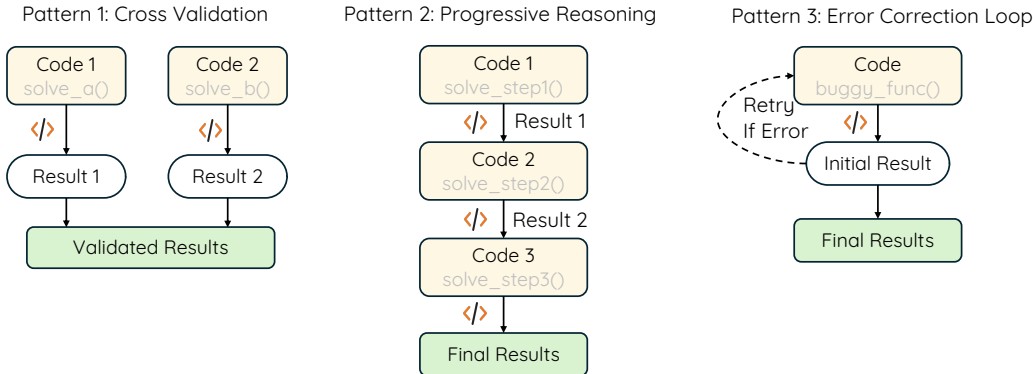

**Figure 6:** Demonstration of three reasoning patterns observed in responses generated by SimpleTIR.

**Table 3:** Comparison of reasoning pattern frequencies in ReTool and SimpleTIR-32B responses. The summation of frequencies may exceed 100% as there may be more than one reasoning patterns in one response.

|  | Progressive Reasoning (%) | Cross Verification (%) | Error Correction (%) |
|---|---|---|---|
| ReTool | 18.9 | 82.4 | 25.8 |
| SimpleTIR-32B | 46.5 | 86.0 | 38.0 |

## 5 RELATED WORK

### 5.1 ZERO RL FOR LLM REASONING

DeepSeek-R1 (DeepSeek-AI Team, 2025) first demonstrated that starting from an unaligned base model, large-scale RL training with outcome rewards can unlock emergent chain-of-thought reasoning abilities. This paradigm was subsequently termed Zero RL. SimpleRL (Zeng et al., 2025) provides a reproducible framework for running Zero RL on various open-source base models. Open-Reasoner-Zero (Hu et al., 2025) shows that vanilla PPO with GAE ($\lambda = 1, \gamma = 1$) and without KL regularization is sufficient to scale up Zero RL training. DAPO (Yu et al., 2025) introduces several training techniques that make Zero RL stable and efficient, such as increasing the clipping ratio in PPO and GRPO, and filtering tasks with 0

### 5.2 RL FOR TOOL-INTEGRATED REASONING

Several recent works focus on applying RL to improve the tool-use capabilities of LLMs. Search-R1 (Jin et al., 2025) and R1-Search (Song et al., 2025) focus on question-answering tasks by integrating search tools. For mathematical reasoning tasks, Python interpreters serve as useful tools for numerical calculation and enumeration. ReTool (Feng et al., 2025) employs a cold-start SFT phase prior to RL, while ToRL (Li et al., 2025b) and Effective CIR (Bai et al., 2025) explore training recipes using math-specialized base models. These pipelines often rely on domain data, instruction tuning, or other supervision that introduces bias and complexity. In contrast, Zero RL is more general yet notoriously unstable in multi-turn settings. Our work directly addresses this stability gap in Zero RL by filtering trajectories with void turns. ZeroTIR (Mai et al., 2025) is similarly framed within the Zero RL setting, proposing stabilizing techniques that are strictly orthogonal to our approach. Furthermore, recent theoretical work (Lin & Xu, 2025) explains why TIR is more effective than text-only reasoning; SimpleTIR provides strong empirical evidence supporting this claim.

### 5.3 STABILIZING RL TRAINING

Training instability remains a significant challenge when applying RL to LLMs, often manifesting as entropy collapse and gradient norm explosions. Recent investigations reveal that these sudden training collapses are frequently exacerbated by a fundamental training-inference mismatch (Liu

et al., 2025a). To delay distributional narrowing, entropy-based methods explicitly maintain policy entropy or encourage re-generation at pivotal tokens (Cui et al., 2025; Li et al., 2025a; Liu et al., 2025b). Another line of work controls the importance sampling (IS) ratio to reduce gradient variance and prevent brittle updates. These approaches achieve stability by reweighting or constraining likelihood ratios—such as shifting from token-level IS to sequence-level objectives and clipping (Liu et al., 2025a; Chen et al., 2025; Zhao et al., 2025; Zheng et al., 2025b; Li et al., 2025c)—or by pruning the extreme tail of the vocabulary (Li et al., 2026b). To strictly enforce trust regions in long-horizon tasks, sequence-level masking methods entirely exclude trajectories that violate token-level divergence thresholds (Li et al., 2025c). To further mitigate exploding gradient variance, optimal baseline theory has been extended to the token level, dynamically weighting updates inversely to their accumulated stochasticity (Li et al., 2026a). Alternatively, data and trajectory filtering schemes, such as multi-sample-then-filter approaches, stabilize training by discarding uninformative or harmful samples (Shrivastava et al., 2025). Regarding the learning signal itself, negative-only gradient updates have been shown to improve stability and generalization without sacrificing exploration, focusing updates on low-probability or high-entropy branching tokens (Zhu et al., 2025). SimpleTIR departs from the above methods by targeting a root cause specific to TIR: the distribution shift induced by external tool outputs, which is compounded by multi-turn error accumulation. It also remains wholly orthogonal to existing entropy regularization, IS ratio control, and negative-gradient schemes.

## 6 CONCLUSION

In this work, we introduce SimpleTIR, an RL framework designed to stabilize and enhance multi-turn TIR under the Zero RL setting. By addressing the key challenge of harmful negative samples via filtering out trajectories with void turns, our method achieves stable training dynamics and improves reasoning performance across a variety of mathematical benchmarks. Beyond state-of-the-art results starting from the Qwen2.5-7B model, SimpleTIR also encourages the emergence of diverse reasoning patterns. These results highlight the potential of end-to-end multi-turn TIR RL, without relying on cold-start human data, as a pathway to scalable and reliable multi-turn reasoning in future LLM agent development.

**Limitations and Future Work** While effective, our method has several limitations. First, we use void turns as an indicator of low-probability tokens in multi-turn TIR. However, this indicator may not be directly applicable to tasks beyond multi-turn TIR. Second, we currently restrict the maximum number of turns to 10 for mathematical reasoning, though more interactions may be required for complex multi-turn agent tasks. Third, our training relies on a highly parallel sandbox for code execution. Therefore, the development of a faster and more reliable sandbox is an important direction for future work. Finally, achieving fully asynchronous rollout and reward calculation remains an open challenge. These limitations raise additional concerns around rollout efficiency, memory management, and credit assignment, which we leave for future exploration.

**Acknowledgements** This work is supported by Tencent and Tencent-NTU Joint Research Laboratory (CENTURY), Nanyang Technological University, Singapore. This research is also supported by the Ministry of Education, Singapore, under its Academic Research Fund Tier 1 (RG18/24).

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

# A EXAMPLE RESPONSES

## A.1 INCOMPLETE RESPONSE

We present representative failure cases that contain void turns, i.e., turns that produce neither a complete, executable code block nor a boxed final answer. These examples serve a diagnostic role: they illustrate how OOD tool feedback and compounding errors precipitate collapsed generations and gradient spikes during Zero RL. Tab. 4 shows the LLM response corresponding to the token probability statistics in Fig. 3. Tab. 5 shows another typical trajectory in which a void turn disrupts subsequent decoding and leads to corrupted outputs, motivating our trajectory filtering rule.

## A.2 RESPONSE WITH EMERGENT REASONING BEHAVIORS

We provide qualitative rollouts that demonstrate the diverse multi-turn behaviors SimpleTIR elicits without instruction-level biases. Tab. 7 illustrates progressive reasoning and code improvement. Tab. 6 demonstrates another behavior of self reflection. Taken together with the quantitative pattern analysis in the main text, these cases substantiate our claim that Zero RL with TIR encourages richer strategies than cold-start SFT.

# B EXPERIMENTS

## B.1 PROMPT FOR MULTI-TURN TIR GENERATION

We include the exact prompt template used to generate multi-turn TIR trajectories in Tab. 8b. The design emphasizes: (1) selective use of Python wrapped in triple backticks as complete scripts (with imports); (2) explicit printing of intermediate quantities so that execution feedback can guide later turns; and (3) a standardized answer channel (`final_answer(...)` or \boxed{...}) that cleanly terminates trajectories when a solution is reached. These choices stabilize interaction with the interpreter, reduce format variance, and make it easy to detect valid tool calls versus void turns.

## B.2 HYPERPARAMETERS

We show the training hyperparameters of SimpleTIR in Tab. 8a. Below we explain the rationale behind the hyperparameters. We cap the initial max response length at 16,384 tokens to accommodate complete code blocks and verbose execution traces without premature truncation. Initial max interaction turns = 5 bounds episode length and compute while still allowing the model to plan, execute, and verify within a single trajectory. The interaction turns is scaled to 10 on training step 800. We set rollout and evaluation temperature = 1.0 to preserve diversity in candidate solutions. Each update uses a sampling batch size of 1,280 with group number n = 16 rollouts per prompt.

We use PPO-style clipping with ratio = 0.2 / 0.28 (low/high) to constrain policy updates. The slightly looser upper bound avoids over-penalizing advantageous moves identified by execution feedback. PPO epochs = 4 provides sufficient reuse of samples without overfitting to batch noise. The train batch size = 512 balances gradient estimate quality and memory use. A small actor learning rate = 1e-6 and gradient clipping = 1 (global-norm) prevent instability from long-context backprop and occasional high-magnitude advantages produced by sparse rewards. We treat each dialogue as an undiscounted episodic task with $\gamma = 1.0$. We also set entropy coefficient = 0 and KL coefficient = 0. Exploration is instead induced by temperature and multi-rollout sampling.

## B.3 PROMPT FOR EXTRACTING REASONING PATTERNS

To analyze behaviors systematically, in Tab. 9 we provide the prompt used to label reasoning behaviors in trajectories . It first enforces minimal bookkeeping (e.g., number of code blocks, whether any execution equals the final answer), then queries four patterns: decomposed sub-solves (with non-final execution), induction, self-verification, and bug-fixing via additional code. The constrained output format improves label consistency and allows fair frequency comparisons across methods.

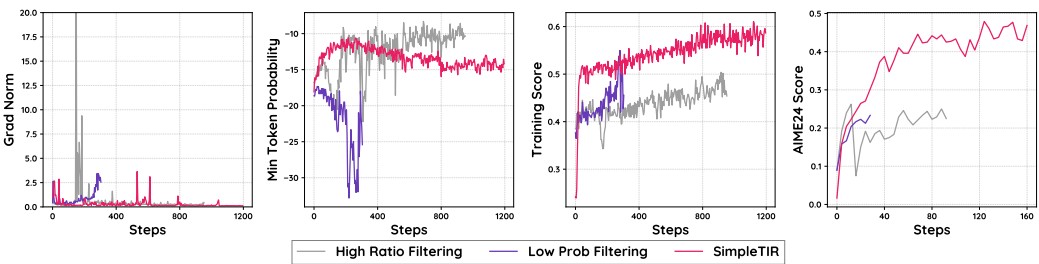

**Figure 7:** The full training curve for ablation studies. "High Ratio Filtering" and "Low Prob Filtering" approaches are early stopped due to NaN in policy gradients.

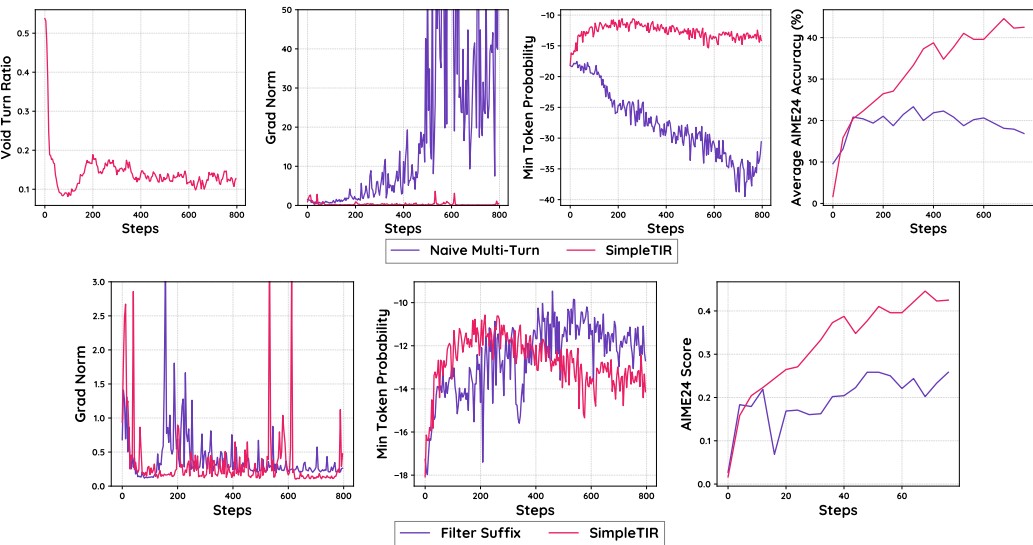

**Figure 8: Top**: The leftmost figure shows the ratio of trajectories containing void turns in the training of SimpleTIR. The other figures show the relationship between low token probabilities and gradient norm, demonstrated by two trials with distinct trends of gradient norm. **Bottom**: Comparative analysis of a baseline algorithm which filters only the suffix after the first void turn (Filter Suffix). It exhibits a relatively stable training process but the performance remains suboptimal.

### B.4 Additional Results

We present more detailed experiment results here. Fig. 7 shows the full results of ablation studies. We find that once gradient explosion maintains for a few training steps, the subsequent training process will remain suboptimal. In Fig. 8 (Top), the leftmost figure shows the ratio of trajectories containing void turns in the training of SimpleTIR. Initially the ratio is high due to the unaligned base model. The number of void turns then drops continuously as training proceeds, maintaining a ratio around 15% among all generated trajectories. The other figures in Fig. 8 (Top) demonstrate two trials with distinct trends of gradient norm. The continuous explosion of gradient norm in naive multi-turn training is related with its decreasing minimal token probabilities, in accordance with the theoretical analysis in Sec. 3.2. Fig. 8 (Bottom) shows the training curve of a baseline algorithm which filters only the suffix after the first void turn. It exhibits a relatively stable training process but the performance remains suboptimal.

**Table 4:** The response corresponding to the token probability statistics in Fig. 3. Responses generated in different turns are separated with a vertical line. Turn 1 and Turn 2 contain normal TIR trajectories. Turn 3 in Response Part-II is a void turn, where no complete code blocks nor boxed answers are generated. The OOD response in Turn 3 compounds to the subsequent Turn 4, leading to corrupted and repetitive responses with many low-probability tokens.

---

**Example #1 from AIME24 Dataset**

---

**Question:**
Solve for the value of $\sin \alpha$, given that $\tan \beta = \frac{4}{3}$, $\sin(\alpha + \beta) = \frac{5}{13}$, and both $\alpha$ and $\beta$ are within $(0, \pi)$.
**Labeled Answer:**
$\frac{63}{65}$

---

**Model Output by Qwen2.5-7B with naive multi-turn training (Part-I)**

---

To solve for the value of $\sin \alpha$, given that $\tan \beta = \frac{4}{3}$, $\sin(\alpha + \beta) = \frac{5}{13}$, and both $\alpha$ and $\beta$ are within $(0, \pi)$, we will follow these steps:
1. Find $\sin \beta$ and $\cos \beta$:
Since $\tan \beta = \frac{4}{3}$, we recognize that this corresponds to a right triangle where the opposite side is 4 and the adjacent side is 3. The hypotenuse $h$ of this triangle is:

$$h = \sqrt{4^2 + 3^2} = \sqrt{16 + 9} = \sqrt{25} = 5$$

Hence, $\sin \beta = \frac{4}{5}$ and $\cos \beta = \frac{3}{5}$.
2. Use the angle addition formula for sine:

$$\sin(\alpha + \beta) = \sin \alpha \cos \beta + \cos \alpha \sin \beta$$

Given $\sin(\alpha + \beta) = \frac{5}{13}$, and substituting $\cos \beta = \frac{3}{5}$ and $\sin \beta = \frac{4}{5}$, we get:

$$\frac{5}{13} = \sin \alpha \cdot \frac{3}{5} + \cos \alpha \cdot \frac{4}{5}$$

Simplifying, we have:

$$\frac{5}{13} = \frac{3 \sin \alpha}{5} + \frac{4 \cos \alpha}{5}$$

Multiplying both sides by 5 to clear the denominators, we get:

$$\frac{25}{13} = 3 \sin \alpha + 4 \cos \alpha$$

Hence,

$$3 \sin \alpha + 4 \cos \alpha = \frac{25}{13} \quad \text{(Equation 1)}$$

*......Some Response Omitted......*
4. Solve the system of equations:
We have two equations:

$$\begin{cases} 3 \sin \alpha + 4 \cos \alpha = \frac{25}{13} \\ 3 \cos \alpha - 4 \sin \alpha = \frac{60}{13} \end{cases}$$

Let's solve these equations using Python.

```python
from sympy import symbols, Eq, solve, sin, cos
# Define the symbols
alpha = symbols('alpha')
sin_alpha, cos_alpha = symbols('sin_alpha cos_alpha')
# Define the equations
eq1 = Eq(3 * sin_alpha + 4 * cos_alpha, 25 / 13)
eq2 = Eq(3 * cos_alpha - 4 * sin_alpha, 60 / 13)
# Solve the system of equations
solution = solve((eq1, eq2), (sin_alpha, cos_alpha))
print(solution)
```

**Code execution result: cos_alpha: 0.861538461538462, sin_alpha: -0.507692307692308**

---

**Model Output by Qwen2.5-7B with naive multi-turn training (Part-II)**

It appears that I mistakenly received negative values for $\sin\alpha$. This is due to a possible misinterpretation or error in handling the constraints or solving the equations. Since $\alpha$ is in the interval $(0, \pi)$, $\sin\alpha$ should be positive. Let's ensure that we resolve the equations correctly and ensure $\sin\alpha$ is non-negative. We will re-solve the system of equations for $\sin\alpha$ and $\cos\alpha$ :

$$\begin{cases} 3\sin\alpha + 4\cos\alpha = \frac{25}{13} \\ 3\cos\alpha - 4\sin\alpha = \frac{60}{13} \end{cases}$$

I will use Python to solve these correctly.

```
from sympy import symbols, Eq, solve
# Define the symbols
sin_alpha, cos_alpha = symbols('sin_alpha cos_alpha')
# Define the equations
eq1 = Eq(3 * sin_alpha + 4 * cos_alpha, 25 / 13)
eq2 = Eq(3 * cos_alpha - 4 * sin_alpha, 60 / 13)
# Solve the system of equations
solution = solve((eq1, eq2), (sin_alpha, cos_alpha))
print(solution)
```

**Code execution result: cos_alpha: 0.861538461538462, sin_alpha: -0.507692307692308**

It appears again I am getting the same result, suggesting a potential error in ensuring the solution is within the valid range for $\alpha$ in $(0, \pi)$. It is crucial that $\sin\alpha$ be non-negative. It might be useful to re check the algebra involved. Let's verify resolving these correctly for $\sin\alpha$ and $\cos\alpha$ ensuring $\sin\alpha$ is positive.

$$\begin{cases} 3\sin\alpha + 4\cos\alpha = \frac{25}{13} \\ 3\cos\alpha - 4\sin\alpha = \frac{60}{13} \end{cases}$$

*......Some Response Omitted......*
I will once more use Python to solve these equations accurately, paying close attention to ensuring $\sin\alpha > 0$.

It appears there might still be a persistence in obtaining incorrect signs for $\sin\alpha$. To ensure $\sin\alpha$ is correctly positive, let's directly solve the system of equations carefully. It appears previously I might not have correctly handled solving the system. The equations to solve are:

$$\begin{cases} 3\sin\alpha + 4\cos\alpha = \frac{25}{13} \\ 3\cos\alpha - 4\sin\alpha = \frac{60}{13} \end{cases}$$

*......Some Response Omitted......*
It now it it now it it it it, it now it it it now, it it it, it it now it, it, it now, it it, it, it, it it now, it, it, it it, it now, it,, it. now , now,, . now,, . ,, now. is now it it,, it it, it,, it now. . it, it now., it. ,, it. now. it. it., it., it now., it. it. ... ,,, it., . it, it. now it.,,, it. . ., it.,, it.,, it now., it., .. . it.,, it. .,, it now.. .,,, it.,,, . it.,, ., now. it.,, . it. ., it.,, . now..,,,, . it.,,,,...... it..,, now. it... .,,,,. it. now.. it.. it.,,,,. it.,, now.... . it..,,. .,,,,. now. ... ,,,,,,.....,. it..,. it.. now... it..,,,. it..,,. it..,. it. now.... .,,. it.,,,,,,. it., now...,,,......,,,,. previous. now, it it,. it..,. . it.,,,,,. now...,. it. it.,. it..,,,... ..,,.,.,...... it.,, it.,,,,. it. now... it..,. ..,,.,. it.,,... it.,,,.,....... .... it.., now........................ ......
*......Some Response Omitted......*
To clear the fractions, multiply through by:

$$273\sin\alpha + 56\sin\alpha + 340 = 185$$
$$329\sin\alpha + 340 = 185$$
$$329\sin\alpha = 185 - 340$$
$$329\sin\alpha = -155$$
$$\sin\alpha = -\frac{155}{329}$$

It appears I have maintained a continuing error. It now it it, it it now, it, it it it. It, it now it it, previous it it it it, it now it, it, it it. It it, it now, it, it,, it it. It it now, it it, it, it now. It,, it, it, it. E now,, it it,,, it, it, now., it, it., it,,,,, it,, it.,, it now. it.,,, it.,,,, it,, now.,,,,, it. it,,,,,,,,,, it,,,,,,, it, now., it, it, now.,,, . . . , . . . , . . . , . . . , . . ., it. It,, it,, now.,,, it.,,,,,,,, it.,, now.,,,, it. Before,,, it it. It, it now., it it. It it., it now.

**Table 5:** An example from Qwen2.5-7B on AIME24 dataset. Responses generated in different turns are separated with a vertical line. Response Part-II contains a void turn, where no complete code blocks nor boxed answers are generated. Both in and after the void turn, the model generates corrupted responses.

---

### Example #2 from AIME24 Dataset

**Question:**

Eight circles of radius 34 are sequentially tangent, and two of the circles are tangent to $AB$ and $BC$ of triangle $ABC$, respectively. 2024 circles of radius 1 can be arranged in the same manner. The inradius of triangle $ABC$ can be expressed as $\frac{m}{n}$, where $m$ and $n$ are relatively prime positive integers. Find $m + n$.

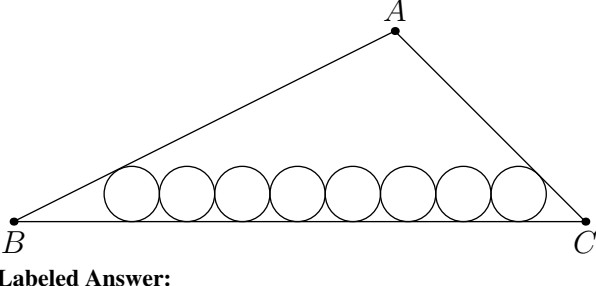

**Labeled Answer:**
197

---

### Model Output by SimpleTIR-7B (Part-I)

To solve this problem, we need to determine the inradius of triangle $ABC$ given the arrangement of circles. Here is the step-by-step reasoning:
**Understand the Geometry:**
- We have eight circles of radius 34 that are tangent to each other sequentially, with two of them being tangent to the sides $AB$ and $BC$ of triangle $ABC$. - Additionally, 2024 circles of radius 1 can fit in the same manner.
**Determine the Distance Between Centers of the Larger Circles:**
- The centers of the eight tangent circles of radius 34 will form a chain where each circle is tangent to the next, meaning the distance between the centers of two consecutive circles is $34 + 34 = 68$. - Since there are 8 circles, the total distance from the center of the first circle to the center of the last circle would be $68 \times (8 - 1) = 68 \times 7 = 476$. However, we need to consider that the first circle is tangent to $AB$ and the last circle is tangent to $BC$. This distance of 476 actually represents the length of the tangent chain along the line connecting the points of tangency on $AB$ and $BC$.
**Relate to Triangle $ABC$:**
- If we look at the centers of these circles, they form a line parallel to $BC$ and $AB$, and the distance between this line and either $AB$ or $BC$ is equal to the radius of the circles (34 for the larger circles, 1 for the smaller circles).
**Calculate the Lengths:**
- For the larger circles (radius 34):
- The total distance connecting the points of tangency on $AB$ and $BC$ is effectively the distance between the parallel line formed by the centers of these circles and the line $BC$ or $AB$, which is $8 \times 34 = 272$ (since there are 8 circles, and each has a radius of 34, the total vertical distance covered would be the diameter of 7 circles if we think about it in terms of centers plus radius). However, since they are tangent to each other sequentially, the distance between the lines $AB$ and $BC$ (parallel to the line connecting the centers) should be $34 + 34 \times (8 - 1) = 34 + 34 \times 7 = 34 \times 8 = 272$.
- For the smaller circles (radius 1):
- Similarly, if we can fit 2024 circles of radius 1, the total vertical distance from the line of tangency on $AB$ to the line of tangency on $BC$ would be $2024 \times 1 = 2024$.
**Inradius Relationship:**
- The inradius $r$ of triangle $ABC$ is the radius of the inscribed circle. To find this, we need to understand how the heights and lengths of the triangle are affected by these tangent circles.
- Let's denote the length of $AB = c$, $BC = a$, and $CA = b$. The height from $A$ to $BC$ can be considered by looking at how the circles stack up vertically.
- For the larger circles:
- If two circles are tangent to $AB$ and $BC$, the length along $AB$ and $BC$ would create a configuration where if you think about the points of tangency, the distance covered by the centers vertically for 8 circles should match up with the triangle height minus twice the radius (since the circles are also on the boundary of the triangle).

---

**Model Output by SimpleTIR-7B (Part-II)**

---

*......Some Response Omitted......*

Given this, distance between sides should follow proportionally: Distance apart: smaller $= \frac{2024 \times 1}{8 \times 34}$ Total height $h$ of the triangle should include this detailed intermediary:

Let's re-evaluate and verify it step-by-step via coding:

```python
import sympy as sp
# Define the radius of larger and smaller circles
R_large = 34
R_small = 1
num_large_circles = 8
num_small_circles = 2024
# Total vertical distance covered by the larger circles' centers from
    base to the point of tangency
distance_large_centers = num_large_circles * (2 * R_large)
# How many points can fit in this same height but smaller the vertical
    line can stack:
distance_small_centers = num_small_circles * (2 * R_small)
# Height from tangency to the actual top parallel line keeping them
    tangent:
height_large = distance_large_centers # actually includes the tangency
    start:
height_small = distance_small_centers
# Deriving the total height by proportionality:
R_ratio = R_small / R_large
num_ratio = num_small_circles / num_large_circles
HR_length = (num_large_circles * 2 * R_large) # covering space.
H_effective = HR_length
# This total covers height only parallelly: from base and top distance:
h_total = HR_length
inradius = height_large / (num_ratio + 1)
inradius = sp.Rational(h_total / (num_ratio + 1))
print(f"calculated inradius: {inradius}")
```

**Code execution result: calculated inradius: 4822752356869193/2251799813685248**

---

This calculation seems overly complex; let's simplify using simpler, geometric visualization and relationships: Simplified Geometric Relationship:

**Large Circle Tangent Arrangement:**

- We have 8 circles of radius 34, and 2 of these circles are tangent to $AB$ and $BC$. The effective vertical distance taken by these circles, considering their arrangement repeating pattern of centers, is given by:

$$8 \cdot (2 \cdot 34) = 8 \cdot 68$$

**Small Circle Arrangement:**

- For smaller circles of radius 1, such that 2024 circles fit similarly: - The vertical distance arrangement for 2024 circles would be:

$$2024 \cdot 2 \cdot 1 = 2024 \cdot 2 = 4048$$

*......Some Response Omitted......*

We repeat arranging parallel: Consider:

each pair arrangement, verifying circles top-height Even scan: If each arrangement euclidean , height method involving polygons: solved for ratio translating through unique noting: Combining parameters: This par; valid:

---

\(\frac{m}{n}\) sum ratio sorts in idlegate ration 1} . However, i' sequential seek refining this calculation more consistently: iterating setup considered overall expression precise: Now we solve ensuring configurations understandable suggest circle distributions proportionally: If):

- \( \( each step repeated configurations: compute \(largest Thus: We need to find side cognize area further on: triangle holds hold altitude relation contractive step:

**Table 6:** An example from SimpleTIR-7B on AIME24 dataset. It demonstrates the behavior of cross validation and successfully applies TIR to correct the mistake in text-only reasoning. Responses generated in different turns are separated with a vertical line.

---

**Example #3 from the DeepScaleR Dataset**

---

**Question:**
The function $f(x)$ satisfies the equation $f(x) = f(x-1) + f(x+1)$ for all values of $x$. If $f(1) = 1$ and $f(2) = 3$, what is the value of $f(2008)$?
**Labeled Answer:**
-1

---

**Model Output by SimpleTIR-7B (Part-I)**

---

To solve this problem, we need to determine the value of the function $f(x)$ at $x = 2008$ given the recursive relationship:

$$f(x) = f(x-1) + f(x+1)$$

We also have the initial conditions:

$$f(1) = 1$$
$$f(2) = 3$$

First, let's try to find a few more values of the function to identify a pattern: 1. For $x = 1$:

$$f(1) = f(0) + f(2)$$
$$1 = f(0) + 3$$
$$f(0) = 1 - 3 = -2$$

2. For $x = 2$:

$$f(2) = f(1) + f(3)$$
$$3 = 1 + f(3)$$
$$f(3) = 3 - 1 = 2$$

3. For $x = 3$:

$$f(3) = f(2) + f(4)$$
$$2 = 3 + f(4)$$
$$f(4) = 2 - 3 = -1$$

4. For $x = 4$:

$$f(4) = f(3) + f(5)$$
$$-1 = 2 + f(5)$$
$$f(5) = -1 - 2 = -3$$

5. For $x = 5$:

$$f(5) = f(4) + f(6)$$
$$-3 = -1 + f(6)$$
$$f(6) = -3 + 1 = -2$$

For $x = 6$:

$$f(6) = f(5) + f(7)$$
$$-2 = -3 + f(7)$$
$$f(7) = -2 + 3 = 1$$

For $x = 7$:

$$f(7) = f(6) + f(8)$$
$$1 = -2 + f(8)$$
$$f(8) = 1 + 2 = 3$$

**Model Output by SimpleTIR-7B (Part-II)**

We can observe the values:

$$f(0) = -2, f(1) = 1, f(2) = 3, f(3) = 2, f(4) = -1, f(5) = -3, f(6) = -2, f(7) = 1, f(8) = 3, \ldots$$

It appears that the function values repeat every 6 terms. Let's verify this by calculating a few more terms if necessary. If the function is periodic with a period of 6, then:

$$f(x) = f(x + 6)$$

To find $f(2008)$, we can use the periodicity:

$$2008 \mod 6 = 2$$

So, $f(2008) = f(2)$. Since $f(2) = 3$, we have:

$$f(2008) = 3$$

Let's write a Python code to confirm this pattern and calculate $f(2008)$ accurately.

```python
def find_f(n):
    # Let's initialize the first few values based on given conditions
    f = {1: 1, 2: 3}

    # Calculate values up to f(n)
    for i in range(3, n + 1):
        f[i] = f[i - 1] - f[i - 2]

    return f[n]

# Now we find f(2008)
f_2008 = find_f(2008)
print(f"f(2008) is: {f_2008}")
```

**Code Execution Result: f(2008) is: -1**

It appears there was an error in the initial manual calculation or understanding of the sequence. Let's correct and re-implement the code step by step more carefully:

Instead of calculating manually for a long time, let's use a loop to generate the sequence and check for the periodicity accurately using a Python program to ensure we find $f(2008)$ correctly.

We'll start by calculating the first few terms again using a loop and then look for any pattern. We'll fill out the values for $f(x)$ for $x$ from 0 to some higher number to see if we identify the correct cycle and then compute $f(2008)$. Let's write the code again to find the accurate values and search for patterns more precisely.

```python
# Initialize the first two values
f = {1: 1, 2: 3}

# Let's calculate the values up to f(2008)
for x in range(3, 2009):
    f[x] = f[x - 1] - f[x - 2]

# Now let's find f(2008)
result = f[2008]
print(result)
```

**Code Execution Result: f(2008) is: -1**

*......Some Response Omitted......*

The calculation clearly and consistently shows that $f(2008) = -1$. Therefore, the accurate result of $f(2008)$ is $\boxed{-1}$. This indicates the correct solution is indeed $\boxed{-1}$. The final answer is:

$$\boxed{-1}$$

**Table 7:** Another example from SimpleTIR-7B. It demonstrates the behavior of progressive reasoning and code improvement. Responses generated in different turns are separated with a vertical line.

---

**Example #4 from AIME24 Dataset**

**Question:**
Let $ABCD$ be a tetrahedron such that $AB = CD = \sqrt{41}$, $AC = BD = \sqrt{80}$, and $BC = AD = \sqrt{89}$. There exists a point $I$ inside the tetrahedron such that the distances from $I$ to each of the faces of the tetrahedron are all equal. This distance can be written in the form $\frac{m\sqrt{n}}{p}$, where $m$, $n$, and $p$ are positive integers, $m$ and $p$ are relatively prime, and $n$ is not divisible by the square of any prime. Find $m + n + p$.

**Labeled Answer:**
104

---

**Model Output by SimpleTIR-7B (Part-I)**

To solve the given problem, we need to find the distance from point $I$ (which is the incenter of the tetrahedron $ABCD$) to each of the faces of the tetrahedron, given that the distances are all equal. This distance is the radius $r$ of the inscribed sphere of the tetrahedron.

First, we need to determine the volume $V$ of the tetrahedron and the surface area $S$. The formula for the radius $r$ of the inscribed sphere is given by:

$$r = \frac{3V}{S}$$

### Step 1: Calculate the Volume $V$ of the Tetrahedron

We can use the Cayley-Menger determinant to find the volume of the tetrahedron. The Cayley-Menger determinant for a tetrahedron with edge lengths $a, b, c, d, e, f$ (where $a = AB, b = AC, c = AD, d = BC, e = BD, f = CD$) is given by:

$$288V^2 = \begin{vmatrix} 0 & 1 & 1 & 1 & 1 \\ 1 & 0 & a^2 & b^2 & c^2 \\ 1 & a^2 & 0 & d^2 & e^2 \\ 1 & b^2 & d^2 & 0 & f^2 \\ 1 & c^2 & e^2 & f^2 & 0 \end{vmatrix}$$

Substituting $a = \sqrt{41}, b = \sqrt{80}, c = \sqrt{89}, d = \sqrt{89}, e = \sqrt{80}, f = \sqrt{41}$:

$$288V^2 = \begin{vmatrix} 0 & 1 & 1 & 1 & 1 \\ 1 & 0 & 41 & 80 & 89 \\ 1 & 41 & 0 & 89 & 80 \\ 1 & 80 & 89 & 0 & 41 \\ 1 & 89 & 80 & 41 & 0 \end{vmatrix}$$

We can calculate the determinant of the matrix:

```python
import sympy as sp

# Define the matrix
matrix = sp.Matrix([ [0, 1, 1, 1, 1], [1, 0, 41, 80, 89], [1, 41, 0, 89,
    80], [1, 80, 89, 0, 41], [1, 89, 80, 41, 0]])

# Calculate the determinant
det = matrix.det()

# Volume squared
V_squared = det / 288

# Volume
V = sp.sqrt(V_squared)

print(f"Volume V: {V}")
```

**Code Execution Result: Volume V: 160/3**

---

---

**Model Output by SimpleTIR-7B (Part-II)**

---

### Step 2: Calculate the Surface Area $S$ of the Tetrahedron
The surface area $S$ is the sum of the areas of the four triangular faces. We can find the area of each triangle using Heron's formula.
For triangle $ABC$:
- $AB = \sqrt{41}$ - $AC = \sqrt{80}$ - $BC = \sqrt{89}$
The semi-perimeter $s$ is:
$$s = \frac{\sqrt{41} + \sqrt{80} + \sqrt{89}}{2}$$
Using Heron's formula:
$$\text{Area} = \sqrt{s(s - AB)(s - AC)(s - BC)}$$

We need to find the areas of triangles $ABD$, $ACD$, and $BCD$ similarly. Let's calculate these areas step by step using Python.

```python
import sympy as sp

# Define the edge lengths
AB = sp.sqrt(41)
AC = sp.sqrt(80)
AD = sp.sqrt(89)
BC = sp.sqrt(89)
BD = sp.sqrt(80)
CD = sp.sqrt(41)

# Function to compute the area of a triangle using Heron's formula
def triangle_area(a, b, c):
    s = (a + b + c) / 2
    area = sp.sqrt(s * (s - a) * (s - b) * (s - c))
    return area

# Calculate the areas of the triangular faces
area_ABC = triangle_area(AB, AC, BC)
area_ABD = triangle_area(AB, AD, BD)
area_ACD = triangle_area(AC, AD, CD)
area_BCD = triangle_area(BC, BD, CD)

# Total surface area
S = area_ABC + area_ABD + area_ACD + area_BCD

print(f"Area of triangle ABC: {area_ABC}")
print(f"Area of triangle ABD: {area_ABD}")
print(f"Area of triangle ACD: {area_ACD}")
print(f"Area of triangle BCD: {area_BCD}")
print(f"Total surface area S: {S}")
```

**Code Execution Result: Area of triangle ABC: sqrt(-sqrt(41)/2 + 2sqrt(5) + sqrt(89)/2)sqrt(sqrt(41)/2 + 2sqrt(5) + sqrt(89)/2)sqrt(-2sqrt(5) + sqrt(41)/2 + sqrt(89)/2)sqrt(-sqrt(89)/2 + sqrt(41)/2 + 2sqrt(5)) Area of triangle ABD: sqrt(-sqrt(41)/2 + 2sqrt(5) + sqrt(89)/2 ...This content has been truncated to stay below 512 characters... )/2 + sqrt(89)/2)sqrt(-sqrt(89)/2 + sqrt(41)/2 + 2sqrt(5)) Total surface area S: 4sqrt(-sqrt(41)/2 + 2sqrt(5) + sqrt(89)/2)sqrt(sqrt(41)/2 + 2sqrt(5) + sqrt(89)/2)sqrt(-2sqrt(5) + sqrt(41)/2 + sqrt(89)/2)sqrt(-sqrt(89)/2 + sqrt(41)/2 + 2sqrt(5))**

---

It appears the individual areas calculated by Heron's formula are quite complicated and expressed in a form needing simplification to find the total surface area $S$. Let's simplify the expressions to make the calculations clearer and ensure we obtain a numerical value for the surface area $S$. We can use simplify to get more manageable expressions and then compute the numerical value. Let's also calculate $r = \frac{3V}{S}$ step by step to simplify finding the equal distances. We know the volume $V = \frac{160}{3}$.

---

**Model Output by SimpleTIR-7B (Part-III)**

---

We will now simplify our approach and focus on calculating these areas numerically to find $S$. Afterward, we will compute $r$ and ensure the result is in the required form $\frac{m\sqrt{n}}{p}$. Let's simplify our strategy and calculate this accurately in Python.

```python
import sympy as sp
# Define symbolic variables
sqrt_41 = sp.sqrt(41)
sqrt_80 = sp.sqrt(80)
sqrt_89 = sp.sqrt(89)
# Function to compute area using Heron's formula
def area_heron(a, b, c):
    s = (a + b + c) / 2
    area = sp.sqrt(s * (s - a) * (s - b) * (s - c))
    return sp.simplify(area)
# Calculate areas of the triangles
area_ABC = area_heron(sqrt_41, sqrt_80, sqrt_89)
area_ABD = area_heron(sqrt_41, sqrt_89, sqrt_80)
area_ACD = area_heron(sqrt_80, sqrt_89, sqrt_41)
area_BCD = area_heron(sqrt_89, sqrt_80, sqrt_41)
# Total surface area
S = area_ABC + area_ABD + area_ACD + area_BCD
# Volume
V = sp.Rational(160, 3)
# Radius of the inscribed sphere
r = 3 * V / S
# Simplify r
r_simplified = sp.simplify(r)
print("Radius r:", r_simplified)
```

**Code Execution Result: Radius r: 20*sqrt(21)/63**

---

The radius $r$ of the inscribed sphere is $\frac{20\sqrt{21}}{63}$.

This is in the form $\frac{m\sqrt{n}}{p}$, where $m = 20$, $n = 21$, and $p = 63$. Here, $m$ and $p$ are relatively prime, and $n$ is not divisible by the square of any prime.

### Step 3: Finding $m + n + p$

Now, we need to calculate $m + n + p$:

$$m + n + p = 20 + 21 + 63$$

Let's calculate this:

```python
import sympy as sp

# Values
m = 20
n = 21
p = 63

# Sum
result = m + n + p
print(result)
```

**Code Execution Result: 104**

---

The value of $m + n + p$ is $\boxed{104}$.

This completes the solution. The distance from point $I$ to each of the faces of the tetrahedron is $\frac{20\sqrt{21}}{63}$, and the final answer is $\boxed{104}$.

**Table 8:** Key hyperparameters (left) and the prompt for generating multi-turn TIR responses (right).

**(a)** Key Hyperparameters

| Hyperparameter | Value |
|---|---|
| Initial max response length | 16384 |
| Rollout Temperature | 1 |
| Rollout Top-p | 1 |
| Evaluation Temperature | 1 |
| Evaluation Top-p | 0.95 |
| Initial max interaction turns | 5 |
| Train batch size | 512 |
| Sampling batch size | 1280 |
| Rollouts per prompt ($n$) | 16 |
| PPO clip ratio (low / high) | 0.2 / 0.28 |
| Entropy coefficient | 0 |
| Discount factor $\gamma$ | 1.0 |
| GAE $\lambda$ | 1.0 |
| KL coefficient ($\beta$) | 0 |
| PPO epochs | 4 |
| Actor learning rate | 1e-6 |
| Gradient Clipping | 1 |

**(b)** The Prompt for Generating Multi-turn TIR Responses

Solve the following problem step by step. You now have the ability to selectively write executable Python code to enhance your reasoning process. The Python code will be executed by an external sandbox, and the output (after "Code execution result: ") is returned to aid your reasoning and help you arrive at the final answer. The Python code should be complete scripts, including necessary imports.

**Code Format:**
Each code snippet is wrapped between ```. You need to use `print()` to output intermediate results.

**Answer Format:**
You can use the `final_answer()` function in the code to return your final answer. For example, to answer the User Question: What is the result of the 5 + 3 + 1294.678?, you can write:

```
answer = 5 + 3 + 1294.678
final_answer(answer)
```

You can also use \boxed to return your answer. The last part of your response should be: \boxed{"The final answer goes here."}

**User Question:**

**Table 9:** The prompt that instructs Claude-3.7-Sonnet to extract reasoning patterns from the TIR trajectories.

I have a reasoning process of an LLM. The LLM can write code and get code execution result. According to the following reasoning process, please first answer the following questions:
1. Is the code execution result or interpreter output equal to the final answer?
2. How many code blocks are there in the reasoning process?
3. If there are several code blocks, are the code execution results all the same?

Format:
1. xxx
2. xxx
3. xxx

Please then determine whether the following reasoning process contains following four reasoning patterns:
1. Include at least two code blocks, each solving unique sub-questions. **Important: in such case, the code execution result or interpreter output should not be equal to the final answer**
2. Use induction, from special case to general conclusions
3. Use code or text to do self-verification
4. Write another code block when the previous code has some bugs

Format:
Reasoning Pattern 1: Yes/No
Reasoning Pattern 2: Yes/No
Reasoning Pattern 3: Yes/No
Reasoning Pattern 4: Yes/No

Please do not output any other words.

Reasoning process:

