# OpenReview forum: "SimpleTIR: End-to-End Reinforcement Learning for Multi-Turn Tool-Integrated Reasoning"
_ICLR.cc/2026/Conference — ICLR 2026 Poster_

### Official Review · Reviewer_eE8Z · 2025-10-27

**Soundness:** 3
**Presentation:** 3
**Contribution:** 3
**Rating:** 6
**Confidence:** 2

**Summary:**

SimpleTIR tackles the instability of multi-turn tool-integrated reasoning under Zero-RL by pinpointing a root cause: the distributional drift from external tool feedback that increases low-probability token sampling and triggers gradient explosions. The paper then stabilizes training with a plug-and-play trajectory filter that drops “void turns”. The paper formalizes the issue with a hierarchical MDP view and a logits-space gradient analysis, then shows strong empirical gains.

**Strengths:**

(1) The paper did sharp diagnosis of a concrete failure mechanismgrounded in theory, not just anecdote;
(2) Proposes a minimal fix that is orthogonal to RL algorithm choice and easy to adopt;
(3) Present consistent empirical wins and smooth, stable training across turn budgets
(4) Present evidence of emergent, diverse reasoning patterns preserved by Zero-RL rather than constrained by SFT “cold starts.”

**Weaknesses:**

(1) The “void-turn” heuristic may miss other harmful trajectories and could be task-specific beyond code-based TIR;

(2) Experimental design requires max turns are capped (≤10) and can be further extended

**Questions:**

Stronger RL training algorithms for tool using like ToolRL for stronger baseline ?

---

> ### Author Response · Authors · 2025-11-22
> **Author Response**
>
> We thank the reviewer for the insightful reviews. We present our responses as follows.
>
> **Q: The “void-turn” heuristic may miss other harmful trajectories and could be task-specific beyond code-based TIR**
>
> A: There may be other criteria, such as high substring repeatness rate or garbled text, so that more harmful trajectories can be filtered. But they would require additional computational resources to detect. We find that the void turn is most efficient and effective. This single criterion is enough to maintain a stable training process for thousands of steps. For the effectiveness of void turn masking in other tasks, we refer the reviewer to common response 4, where we show the experiment results in QA tasks with search engines as the tool. SimpleTIR also achieves better performance compared with the naive GRPO algorithm.
>
> **Q: Experimental design requires max turns are capped (≤10) and can be further extended**
>
> A: We actually conducted experiments with max turns=20. This does not further scale the validation performance and is extremely time-consuming. The flattened performance can be attributed to the tasks: few mathematical problems in the Math and DeepScaler dataset require a large amount of tool calling (more than 10 times) before coming up with the final answer. We leave scaling the number of tool interaction turns in more complex tasks to future work.

---

### Official Review · Reviewer_7iMa · 2025-10-29

**Soundness:** 4
**Presentation:** 3
**Contribution:** 4
**Rating:** 8
**Confidence:** 4

**Summary:**

This paper identifies and mitigates a key failure mode in multi-turn tool-integrated reinforcement learning under the Zero-RL paradigm.
This paper proposes SimpleTIR, a lightweight, plug‑and‑play trajectory filter that drops any trajectory containing a turn that produces neither a complete code block nor a final answer.
Comprehensive experiments on math-reasoning benchmarks (AIME24/25, MATH500, AMC23, HMMT) show that this paper substantially improves both stability and performance.

**Strengths:**

1. The failure‑mode analysis for multi‑turn TIR under Zero RL is clear.
2. The “void‑turn” trajectory filter is easy to implement, orthogonal to PPO/GRPO.
3. Hyperparameters, prompt formats, and pattern‑labeling prompts are clearly specified.
4. Originality is good. Even trajectory/data filtering is not new in RL/RLHF, but the turn‑level “void‑turn” filter tailored to multi‑turn TIR is a useful.

**Weaknesses:**

1. All experiments use a Python interpreter for math. It remains unclear how the void‑turn heuristic transfers to other tool types (search, multi‑API plans), modalities, or noisier settings. Section 6 acknowledges this, but the paper would benefit from even a small study on search‑augmented QA (e.g., a Search‑R1 setup).
2. The definition & detection seem heuristic. “Void turn = neither a complete code block nor a final answer.” In practice, “complete” relies on formatting and the prescribed prompt conventions. The method’s robustness to format drift (e.g., unboxed answers, partial but executable code, multi‑tool calls) is not evaluated. Clarify detection rules and report false‑positive/false‑negative rates.
3. Dropping entire trajectories wastes rollout budget; the paper does not quantify the filter rate over training or its impact on sample efficiency. Reporting the percentage of filtered trajectories vs. steps and wall‑clock/throughput would strengthen the claim.

**Questions:**

1. Notation in sec 2.1 could be clearer (define H, q, K, L explicitly when introducing the hierarchical MDP).
2. The abstract claims “AIME24 from 22.1 to 50.5 (7B)”, but 22.1 does not appear in Table 1; please reconcile and cite the exact baseline used.
3. Figure caption placement/style is inconsistent (some above, some below).
4. What is the wall‑clock overhead introduced by filtering (including sandbox latency)? Any guidance on batch sizes/turn caps (Figure 5 top) under a fixed compute budget?

---

> ### Author Response · Authors · 2025-11-22
> **Author Response**
>
> We thank the reviewer for the insightful reviews. We present our responses as follows.
>
> **Q: Position of figure captions**
>
> A: We thank the reviewer for pointing this out. We have revised the paper and made sure all figure captions appear below the figure.
>
> **Q: The paper would benefit from a study on search‑augmented QA**
>
> A: We refer the reviewer to general response 4, where we show the experiment results in QA tasks with search engines as the tool. SimpleTIR also achieves better performance compared with the naive GRPO algorithm.
>
> **Q: Clarify detection rules and report false‑positive/false‑negative rates.**
>
> A: The detection rule of void turns is straightforward: For each generation turn, we detect whether there is a complete code block or a final answer surrounded by "\boxed". If not, such turns will be identified as void turns. We refer the reviewer to general response 3 for the false positive rate. In general, the false positive rate of void turn detection is very low, making it a good indicator of model output collapse. For false-negative samples, there may be other criteria, such as high substring repeatness rate or garbled text, so that more problematic trajectories can be filtered. But they would require additional computational resources to detect. We find that the void turn is most efficient and effective. This single criterion is enough to maintain a stable training process for thousands of steps.
>
> **Q: Percentage of filtered trajectories**
>
> A: We refer the reviewer to general response 2 for the filter rate. For most of the training process, trajectories containing void turns take up ~15% of all trajectories. As the proportion is relatively low, we do not conduct additional oversampling to the RL rollout process, leaving no extra wall-clock overhead.
>
> **Q: The baseline score 22.1 does not appear in Table 1; please reconcile and cite the exact baseline used.**
>
> A: Thanks for pointing it out. 22.1 is the result of the DAPO algorithm, trained on Qwen2.5-7B model. We add this in the revised paper.
>
> **Q: Wall‑clock overhead introduced by filtering**
>
> A: Void turn masking adds very little wall-clock overhead. We use a simple if statement to determine the existence of void turns. To filter void turns out of policy loss computation, we set the loss mask of trajectories containing void turns to 0, making no revisions to the original training pipeline.
>
> **Q: Guidance on batch sizes/turn caps**
>
> A: We did not tune the batch size and set ppo mini batch size to 128 following baseline algorithms. The ppo mini steps is set to 4 to maintain a moderate degree of off-policiness when performing policy gradients, leading to a batch size of 512. For the number of turns, the models initially consume an average of two generation turns. We set the maximum generation turns to 5, so that most generations will not be prematurely terminated. As training proceeds, we observe that the average number of turns increases, so the maximum generation turns is raised to 10.

---

### Official Review · Reviewer_JcTa · 2025-10-31

**Soundness:** 3
**Presentation:** 3
**Contribution:** 3
**Rating:** 8
**Confidence:** 5

**Summary:**

This paper introduces SimpleTIR, a simple yet effective reinforcement learning (RL) algorithm designed to stabilize multi-turn tool-integrated reasoning (TIR) under the Zero RL setting. The authors identify that training instability in multi-turn TIR arises from “void turns, i.e., turns that produce neither a code block nor a final answer, which cause gradient explosions through compounding low-probability tokens. SimpleTIR addresses this by filtering out entire trajectories containing such void turns during policy updates. Despite its simplicity, this plug-and-play strategy substantially improves training stability and performance across math reasoning benchmarks, achieving promising results starting from base Qwen models and enabling the emergence of diverse multi-turn reasoning behaviors.

**Strengths:**

- The proposed method is simple and surprisingly effective.
The idea of filtering out trajectories with “void turns” is conceptually straightforward yet powerful. It mitigates a key failure mode in multi-turn RL training without introducing architectural or algorithmic complexity, making it easily adoptable in other setups. These simple but effective tricks will be very elegant in practical use.

- The work tackles an increasingly important and difficult problem.
Multi-turn RL for reasoning and tool use is becoming a central direction in scaling LLM capabilities. Addressing training instability in this regime is both timely and practically impactful.

- Empirical results are strong and convincing.
SimpleTIR not only stabilizes training but also significantly improves final accuracy across multiple benchmarks.

- The analysis and visualization are thorough and insightful.
The authors provide solid empirical and theoretical analysis to identify the cause of instability, supported by clear visualizations of gradient dynamics and token probability shifts. These figures convincingly demonstrate how SimpleTIR mitigates the problem.

**Weaknesses:**

- First of all: The figure captions, which are all placed above the figures, violate the conference formatting requirements.

- The paper heavily focuses on “void turns” but fails to illustrate them clearly in the main text.
While the appendix includes qualitative cases, the main body lacks a concrete or conceptual example of what a void turn actually looks like. Without this, readers cannot easily form an intuitive understanding of the concept or of why such turns “make no progress” in reasoning. Explicit examples or snippets in Section 3.3 would make the paper much clearer.

- Discarding trajectories may lead to data inefficiency in sample-scarce settings.
While filtering out void-turn trajectories works well for large-scale math reasoning tasks, it could be wasteful in domains where data collection is expensive. Discussing potential mitigations, such as partial reuse or weighting instead of full removal, would strengthen the paper’s generality. I'm curious if the authors tried other approaches that can preserve the data.

**Questions:**

see above

---

> ### Author Response · Authors · 2025-11-22
> **Author Response**
>
> We thank the reviewer for the insightful reviews. We present our responses as follows.
>
> **Q: Position of figure captions**
>
> A:  We thank the reviewer for pointing this out. We have revised the paper and made sure all figure captions appear below the figure.
>
> **Q: Illustrate void turns in the main paper**
>
> A: Thanks for pointing this out. Unfortunately the space in the main paper is limited and not available for a complete trajectory containing void turns. In the main paper we mention a common cause of void turns: incomplete code blocks, as an intuitive example. We also add a reference to relevant sections in the appendix.
>
> **Q: Alternative approaches that can preserve the data containing void turns**
>
> A: We tried some alternative approaches similar to what Reviewer Pu3v mentioned, such as using a small loss weight on void turns and filtering subsequent turns after the void turn. The first fails due to the exponential increase of gradient norm and decrease of minimum token probabilities, as shown by the gradient curve in Fig. 8 (Top) in Appendix C.4 of the revised paper. The second also fails, as shown in Fig. 8 (Bottom), due to a credit assignment issue. This is because trajectories containing void turns largely receive negative rewards. This overly punishes multi-turn TIR behaviors, leading to a decrease in average turns and overall performance.
>
> Meanwhile, we find that for most of the training process, trajectories containing void turns only take up ~15% of all trajectories. As the proportion is relatively low, currently we do not conduct additional oversampling to the RL rollout process, leaving no extra wall-clock overhead.

---

> > ### Comment · Reviewer_JcTa · 2025-11-28
> > **Thanks for the response**
> >
> > Thanks authors for the response. My concerns are solved and I would recommend authors to incorporate the clarification in the main paper using the extra page. I am happy to maintain the original positive score.

---

> > > ### Author Response · Authors · 2025-11-30
> > > **Author Response**
> > >
> > > We thank the reviewer for the positive feedback. We will use the extra page for illustrations of void turns in the next version of the paper.

---

### Official Review · Reviewer_Pu3v · 2025-11-01

**Soundness:** 3
**Presentation:** 4
**Contribution:** 3
**Rating:** 6
**Confidence:** 3

**Summary:**

This paper introduces SimpleTIR, a plug-and-play method to stabilize multi-turn tool integration reinforcement learning. The authors note that tool feedback can induce distribution shifts and lead to low-probability tokens and gradient spikes during multi-turn iterations. They analyse the root cause of this phenomenon and address it by filtering complete trajectories containing void turns, i.e., turns that do not contain a code block or yield an answer. On maths benchmarks, SimpleTIR significantly enhances the performance of Qwen2.5-7B and larger models, yielding smoother training curves compared to standard multi-turn RL. The study also observes emerging patterns such as cross-validation and error-correction mechanisms.

**Strengths:**

This paper links multi-turn instability to extremely low-probability tokens emerging after model processing tools, demonstrating that these tokens trigger gradient peaks. Proposition 1 and the case study elucidate this mechanism.

This paper proposes a straightforward solution: directly filtering trajectories containing invalid rounds, thereby addressing unstable updates. The algorithm description and mask selection are clear and unambiguous.

Table 1 shows consistent improvements across AIME24, AIME25, Math500, AMC23 and more, for 7B, 32B and 4B bases, compared to non-TIR RL baselines and prior TIR systems.

SimpleTIR exhibits smooth training curves. Ablation experiments reveal that filtering mechanisms based on low token probability or high ratio demonstrate poor stability. Similarly, filtering only “stop generation instructions” without loss masking yields suboptimal results.

**Weaknesses:**

Void turns are defined as those where the model fails to generate a complete code block or final answer. The training pipeline detects such instances by enforcing a final answer function and fixed tool output prefixes. The authors also note that this metric may not extend beyond the current multi-turn TIR environment.

All tasks were constrained on math problem-solving using Python tools. The paper itself acknowledges that this metric may only be applicable to multi-turn TIR scenarios. More complex agency tasks potentially require additional turns.

Ablation analysis focuses solely on early training (320 steps) and gives the conclusion that SimpleTIR suffers less from low probability tokens and gradient explosion. It does not mention whether the trajectory filtering with high importance ratios or low probability tokens can resolve the challenge of training instability in the long run.

Filtering the whole trajectory when any void turn appears can reject hard trials and change the reward distribution. There is no quantitative report of filter rate or false positives. It may lean towards keeping easy success trajectories and discard hard trajectories.

The gradient-norm proposition is insightful, but there is no broad quantitative study that correlates measured low-probability token rates with gradient spikes across many runs and seeds. Evidence is mainly a case study and a few curves.

**Questions:**

What fraction of trajectories are filtered in each step? Did you count the false-positives, e.g. a useful reasoning turn get discarded, because no explicitly final answer was provided.

Can you test your method’s generality on different tasks, for example, QA, general knowledge, and use different tools, e.g. web search)?
Your ablations focus only on early steps (first 320 steps) to show instability of trajectory filtering with high importance ratios or low probability tokens. Can you report full training curves to confirm that they do not recover in the long run?

Can you give a correlation between low-probability token segments and gradient spikes over many runs and report statistics instead of only a single case?

Did you try to decrease the weight of late bad turns instead of dropping the whole trajectory? Did you try to filter only the suffix after the first void turn?

---

> ### Author Response · Authors · 2025-11-22
> **Author Response**
>
> We thank the reviewer for the insightful reviews. We present our responses as follows.
>
> **Q: All tasks were constrained on math problem-solving using Python tools.**
>
> A: We refer the reviewer to general response 4, where we show the experiment results in QA tasks with search engines as the tool. SimpleTIR also achieves better performance compared with the naive GRPO algorithm.
>
> **Q: Ablation studies in the long run**
>
> A: We have posted the full ablation studies in Figure 7 in Appendix C.4 of the revised paper. We find that once gradient explosion maintains for a few training steps, the subsequent training process will remain suboptimal. Meanwhile, baseline algorithms often witness gradient NaNs, leading to an early stop of training. So it is unlikely that baseline algorithms can resolve the instability challenge in the long run.
>
> **Q: Quantitative report of filter rate or false positives**
>
> A: We refer the reviewer to general response 2 for the filter rate and general response 3 for the false positive rate. For most of the training process, trajectories containing void turns take up ~15% of all trajectories. As the proportion is relatively low, we do not conduct additional oversampling to the RL rollout process, leaving no extra wall-clock overhead. Meanwhile,  the false positive rate of void turn masking is very low. Therefore, the void turn is a good indicator of model output collapse.
>
> **Q: Quantitative study that correlates measured low-probability token rates with gradient spikes**
>
> A: We compare the training curves of gradient norm and the minimum token probabilities in Figure 7 and Figure 8 (Top) in Appendix C.4 of the revised paper. A naive trial has continuously decreasing minimum token probabilities and the gradient norm simultaneously explodes. Meanwhile, SimpleTIR has relatively stable gradient norm and minimum token probabilities. This indicates a direct correlation between gradient spikes and low-probability tokens, in accordance with the theoretical studies.
>
> **Q: Did you try to decrease the weight of late bad turns instead of dropping the whole trajectory?**
>
> A: We tried this approach and find that with the extra weight as small as 1e-3, late bad turns will still cause catastrophic gradient explosion. This is due to an exponential increase of gradient norm and decrease of minimum token probabilities, as shown in Figure 8 (Top).
>
> **Q: Did you try to filter only the suffix after the first void turn?**
>
> A: We show the training curve for only filtering the suffix after void turns in Fig. 8 (Bottom) in Appendix C.4 of the revised paper. Training will not collapse but turns out to be suboptimal compared with SimpleTIR. This is because trajectories containing void turns largely receive negative rewards. This overly punishes multi-turn TIR behaviors, leading to a decrease in average turns and overall performance. In other words, if we do not mask the whole trajectory containing void turns, the policy loss will lead to a credit assignment issue on multi-turn TIR.

---

> > ### Comment · Reviewer_Pu3v · 2025-11-27
> >
> > Thanks for the detailed response. I would like to increase my score given the solid added experiments.

---

> > > ### Author Response · Authors · 2025-11-30
> > > **Author Response**
> > >
> > > We thank the reviewer for the positive feedback!

---

### Author Response · Authors · 2025-11-22
**General Response**

Dear Reviewers,

We are appreciated for your reviews. We find some concerns are common and present our responses here.

1. **Position of figure captions**: We thank the reviewers for pointing this out. We have revised the paper and made sure all figure captions appear below the figure.
2. **Proportion of trajectories including void turns**: In Figure 8 in Appendix C.4, we demonstrate the training dynamics of the proportion of such trajectories. For most of the training process, trajectories containing void turns take up ~15% of all trajectories. As the proportion is relatively low, we do not conduct additional oversampling to the RL rollout process, leaving no extra wall-clock overhead.
3. **False positive rate of void turn masking**: We define false positive samples as void turns that stop generation normally, such as ending with "Let's write some Python code to solve the issue." However, we find that very few void turns, if not none of them, exhibit such normal terminating behavior. All 30 of our sampled void turns have corrupted outputs or terminate abruptly. These turns can terminate with:

> We can find \\( r \\) and \\( s \\) using these two equations: \\[ r = \\frac{24 + 12\\sqrt{3}}{2} =[eos]

> However, this does not match the desired form \\( \\frac{m\\sqrt{n} - p}{q} \\). Let’s consider that \\( \\sqrt{15713[eos]

> Now, find \\(K\\) and \\(L\\): 1. **Coordinates of \\(K\\):[eos]

> 041 user SyntaxError: invalid syntax Can you check the code part?[eos]

> \\[ D = \\frac{[eos]

> AP = \\frac{2 \\times 5^2 \\times[eos]

> \\angle HJF = \\angle HJF = \\angle HJF = \\angle HJF = \\angle HJF = \\angle HJF[eos]

> Express \\( 468.75 \\) in the form \\( a\\sqrt{b} \\): \\[ 468.75 = \\frac{1[eos]

Through these analysis, we can conclude that the false positive rate of void turn masking is very low. Therefore, the void turn is a good indicator of model output collapse.

4. **Verifying SimpleTIR in more tasks**: We conduct additional experiments with the database search tool and QA datasets. Due to time constraints during rebuttal, 1.5B models training are only carried out for 120 gradient steps and 7B models for 50 steps. As shown in the following table, SimpleTIR achieves better performance compared with the Search-R1 algorithm without void turn masking. The improvement is not as considerable as that with the python interpreter and math datasets. This can be attributed to a smaller degree of output OOD  of the search tool compared with the python interpreter, and the naive multi-turn RL training will not completely crash.

| Model                 | Algorithm | Gradient Steps | NQ   | TriviaQA | PopQA | HotpotQA | 2wiki | Musique | Avg      |
| --------------------- | --------- | -------------- | ---- | -------- | ----- | -------- | ----- | ------- | -------- |
| Qwen2.5-1.5B-Instruct | SimpleTIR | 120            | 51.3 | 44.6     | 57.1  | 21.9     | 28.9  | 15.0    | **36.5** |
| Qwen2.5-1.5B-Instruct | Search-R1 | 120            | 50.0 | 42.9     | 57.1  | 19.3     | 12.5  | 0       | 30.3     |
| Qwen2.5-7B            | SimpleTIR | 50             | 50.0 | 53.6     | 56.7  | 25.0     | 35.9  | 20.0    | **40.2** |
| Qwen2.5-7B            | Search-R1      | 50             | 50.0 | 55.4     | 53.4  | 24.0     | 33.6  | 20.0    | 39.4     |

---

### Meta-Review · Area_Chair_f6vE · 2025-12-09

**Summary:**

This paper proposes a simple yet effective void turn filtering technique to mitigate instability issues in multi-round TIR. Three reviewers agreed to accept the paper, while the response from the fourth reviewer was also positive. The authors submitted a rebuttal addressing most of the reviewers' concerns. The AC agreed to accept the paper.

**Reviewer Concerns:**

Reviewers Pu3v and JcTa responded that their concerns had been addressed, and their scores were adjusted accordingly. Reviewers 7iMa and eE8Z did not participate in the discussion. After the AC read the authors' rebuttal, the AC concluded that most of these reviewers' concerns had been resolved.

**Reviewer Scores:**

Reviewers 7iMa and eE8Z did not participate in the discussion. AC believes that had they participated, they would have maintained their initial positive scores.

---

### Decision · Program_Chairs · 2026-01-26

Accept (Poster)